# HiCImpute: A Bayesian hierarchical model for identifying structural zeros and enhancing single cell Hi-C data

Qing Xie[1], Chenggong Han[1], Victor Jin[2], Shili Lin[1,3,4]*

**1** Interdisciplinary Ph.D. Program in Biostatistics, Ohio State University, Columbus, Ohio, United State of America, **2** Department of Molecular Medicine, University of Texas Health Science Center, San Antonio, Texas, United State of America, **3** Department of Statistics, Ohio State University, Columbus, Ohio, United State of America, **4** Translational Data Analytics Institute, Ohio State University, Columbus, Ohio, United State of America

* shili@stat.osu.edu

**Data Availability Statement:** The HiCImpute R package, together with the prepared R data for the real and simulated data used in this study, are available on Github: https://github.com/sl-lin/HiCImpute.

## Abstract

Single cell Hi-C techniques enable one to study cell to cell variability in chromatin interactions. However, single cell Hi-C (scHi-C) data suffer severely from sparsity, that is, the existence of excess zeros due to insufficient sequencing depth. Complicating the matter further is the fact that not all zeros are created equal: some are due to loci truly not interacting because of the underlying biological mechanism (structural zeros); others are indeed due to insufficient sequencing depth (sampling zeros or dropouts), especially for loci that interact infrequently. Differentiating between structural zeros and dropouts is important since correct inference would improve downstream analyses such as clustering and discovery of subtypes. Nevertheless, distinguishing between these two types of zeros has received little attention in the single cell Hi-C literature, where the issue of sparsity has been addressed mainly as a data quality improvement problem. To fill this gap, in this paper, we propose HiCImpute, a Bayesian hierarchical model that goes beyond data quality improvement by also identifying observed zeros that are in fact structural zeros. HiCImpute takes spatial dependencies of scHi-C 2D data structure into account while also borrowing information from similar single cells and bulk data, when such are available. Through an extensive set of analyses of synthetic and real data, we demonstrate the ability of HiCImpute for identifying structural zeros with high sensitivity, and for accurate imputation of dropout values. Downstream analyses using data improved from HiCImpute yielded much more accurate clustering of cell types compared to using observed data or data improved by several comparison methods. Most significantly, HiCImpute-improved data have led to the identification of subtypes within each of the excitatory neuronal cells of L4 and L5 in the prefrontal cortex.

## Author summary

Single cell Hi-C techniques enable one to study cell to cell variability in chromatin interactions, which has significant implications in gene regulations. However, insufficient

**Funding:** This research was supported in part by a grant from the National Institute of Health R01GM114142 (to SL and VJ). The funder had no role in study design, data collection and analysis, decision to publish, or preparation of the manuscript.

**Competing interests:** The authors have declared that no competing interests exist.

sequencing depth—leading to some chromatin interactions with low frequencies not observed—has resulted in many zeros, called dropouts. There are also zeros due to biological mechanisms rather than insufficient coverage, referred to as structural zeros. As such, dropouts and structural zeros are confounded; that is, observed zeros are a mixture of both types. Differentiating between structural zeros and dropouts is important for improved downstream analyses, including cells-subtype discovery, but there is a paucity of available methods. In this paper, we develop a powerful method, HiCImpute, for identifying structural zeros and imputing dropouts. Through an extensive simulation study, we demonstrate the ability of HiCImpute for identifying structural zeros with high sensitivity and accurate imputation of dropout values, under a variety of settings. Applications of HiCImpute to three datasets yield improved data that lead to more accurate clustering of cell types, and further, discovery of subtypes in two of the cell types in the prefrontal cortex data.

## Introduction

Understanding three-dimensional (3D) chromosome structures and chromatin interactions is essential for interpreting functions of the genome because the spatial organization of a genome plays an important role in gene regulation and maintenance of genome stability [1]. Biochemical methods such as high-throughput chromosome conformation capture coupled with next generation sequencing technology (e.g., Hi-C) provide genome-wide maps of contact frequencies, a proxy for how often any given pair of loci interact in the cell nucleus, the natural 3D space where the chromosomes reside [2]. Bulk Hi-C is an averaged snapshot of millions of cells with limited information on heterogeneity or variability between individual cells. In contrast, single-cell Hi-C (scHi-C) data enable one to construct whole genome structures for single cells, ascertain cell-to-cell variability, and cluster single cells. Such studies can lead to understanding of cell-population compositions and heterogeneity, and has the potential to identify and characterize rare cell populations or cell subtypes in a heterogeneous population [3].

Sparsity is one of the major difficulties in analyzing single cell data, and it is even more challenging for scHi-C data, as sparsity is an order of magnitude more severe compared to most of other types of single-cell data [4]. Since Hi-C data are represented as two-dimensional (2D) contact matrices, the coverage of scHi-C (0.25–1%) is much smaller than that of single cell RNA-seq (scRNA-seq, 5–10%) [4]. A further complication is that, among observed zeros in an scHi-C contact matrix, some are true zeros (i.e. structural zeros—SZs) because the corresponding pairs do not interact with each other at all due to the underlying biological function, whereas others are sampling zeros (i.e., dropouts—DOs) as a result of low sequencing depth. Telling SZs and DOs apart is important as it would improve downstream analysis such as clustering and 3D structure recapitulation. For example, methods for reconstructing 3D structures have included a penalty term to position two loci in the 3D space as far as possible if they do not interact [5, 6]. If there is not sufficient sequencing depth, especially in single cells, and if observed zeros are not correctly identified as SZs and DOs, then, applying such a penalty can lead to an artificial separation of two loci that in fact have coordinated effects on certain biological functions.

Currently, the concepts of SZs and DOs are well understood and have received considerable attention in scRNA-seq research, with a number of methods developed to identify SZs and impute DOs. Several of the methods, including MAGIC [7], SAVER [8], scUnif [9], scImpute [10], MCImpute [11], and DrImpute [12], were evaluated and compared in a recent

publication [13]. In contrast, the concepts of SZs and DOs have not been widely pursued in scHi-C research. In fact, although the issue of sparsity has been addressed, albeit still quite limited, in the scHi-C or bulk Hi-C literature, the focus has been on improving data quality, and little has been said about distinguishing between SZs and DOs [14]. Nevertheless, the need for imputing the zeros have been emphasized in several papers, which is treated as a necessary intermediate steps in these papers to improve data quality for answering various biological questions, including assessing data reproducibility, enhancing data resolution, constructing 3D structure, and clustering of single cells [4, 15–18].

Existing approaches for addressing sparsity to improve data quality all aim to "smooth" the data by borrowing information from neighbors, and they may be classified into three categories depending on the methodology used: (1) kernel smoothing, (2) random walks, and (3) convolutional neural network, with representatives in all categories provided in S1 Table. Although neighborhood information in the same cell is used, none of these methods further utilize information from similar single cells or bulk data when they are available. For kernel smoothing, the types of kernels that have been used in the literature are uniform kernels or 2D Gaussian kernels [16]. For example, HiCRep [15], which aims to assess the reproducibility of Hi-C data, applies a uniform kernel (or referred to as 2D mean filters in that paper) by replacing each entry in the 2D contact matrix with the mean count of all contacts in a neighborhood. Another method, scHiCluster [4], has proposed the use of a method in its first step that may also be classified into this category: it uses a filter that is equivalent to taking the average of the genomic neighbors, although the filter may also incorporate different weights during imputation. While a uniform kernel (2D mean filter) takes the average of the genomic neighbors with equal weights, a 2D Gaussian kernel uses a weighted average of neighboring counts according to a 2D Gaussian distribution: the farther away a neighbor is from the entry that is being imputed, the smaller the weight. For instance, SCL [16] applies a 2D Gaussian function to impute scHi-C contact matrices before inferring the 3D chromosome structure.

Method referred to as random walks have also been proposed as a way to smooth out an observed 2D matrix for improving data quality [4, 17, 19, 20]. The idea of a "random walk" process is to borrow information from neighbors in a fashion different from the "neighborhood" idea in kernel smoothing. Any position that is on the same row or column as the entry being imputed (but not necessarily has to be a neighbor) will contribute to the "smoothed" count in each step of the random walk. In GenomeDISCO [17], it is found that taking three steps of the random walk would lead to the best results in the problems investigated therein. Another way to improve data quality is through applying convolutional neural network, a deep learning method commonly applied to analyzing imaging data; HiCPlus [18] and DeepHiC [21] are such supervised learning techniques for improving data quality.

In this paper, we address the following challenging problems: separating the zeros into structural zeros and dropouts, imputing those that are dropouts, and improving data quality more generally. We develop HiCImpute, a Bayesian hierarchical model for single cell Hi-C data that borrows information from three sources (if available): neighborhood of a position in the 2D matrix, similar single cells, and bulk data. The Bayesian hierarchical model facilitates a multi-level modeling approach to integrate different sources of information for making inference about SZ status and imputing DOs. This paradigm allows us to use not only scHi-C count data, but also the above-mentioned three additional pieces of information through the prior component. The information for these priors is further strung together through hierarchical modeling to borrow information from one another to achieve greater consistency. Since counts as well as SZ/DO status in a 2D matrix are correlated with one another, the Bayesian hierarchical framework is particularly suited in this situation. Through an extensive set of analyses of synthetic and real data, we evaluated the ability of HiCImpute for identifying SZs, its

accuracy for imputing DO values, and compare the performance with three existing methods for data quality improvement. We further evaluate downstream analyses using data improved from HiCImpute and the other methods to evaluate the improvement for cell type clustering and subtype discovery.

## Results

### Overview of HiCImpute

The overall goal of HiCImpute, a Bayesian hierarchical model for analyzing single cell Hi-C data, is to identify SZs with high sensitivity and to impute DO values with great accuracy (Fig 1). The key idea relies on the introduction of an indicator variable (the latent variable) denoting SZ or otherwise, for which a statistical inference is made based on its posterior probability estimated using Markov chain Monte Carlo (MCMC) samples (see Methods). We further include additional information through hierarchical modeling. That is, in addition to the model for the data and latent variable, we also specify prior and hyperprior distributions for information from several sources such as neighborhood, similar single cells, and bulk data. We balance the two competing goals—identifying SZs while also accurately imputing DOs—by factoring both into the Bayesian modeling so that they are parameters of interest in the posterior distribution and statistical inference (Output module in Fig 1). A number of criteria for evaluating the performances of HiCImpute have been devised (See Methods). Briefly, the criteria include the proportion of true structural zeros (PTSZ) correctly identified to ascertain the power (sensitivity) for detecting true structural zeros, the proportion of true dropouts (PTDO) identified to gauge the ability for correct identification of dropout events (specificity),

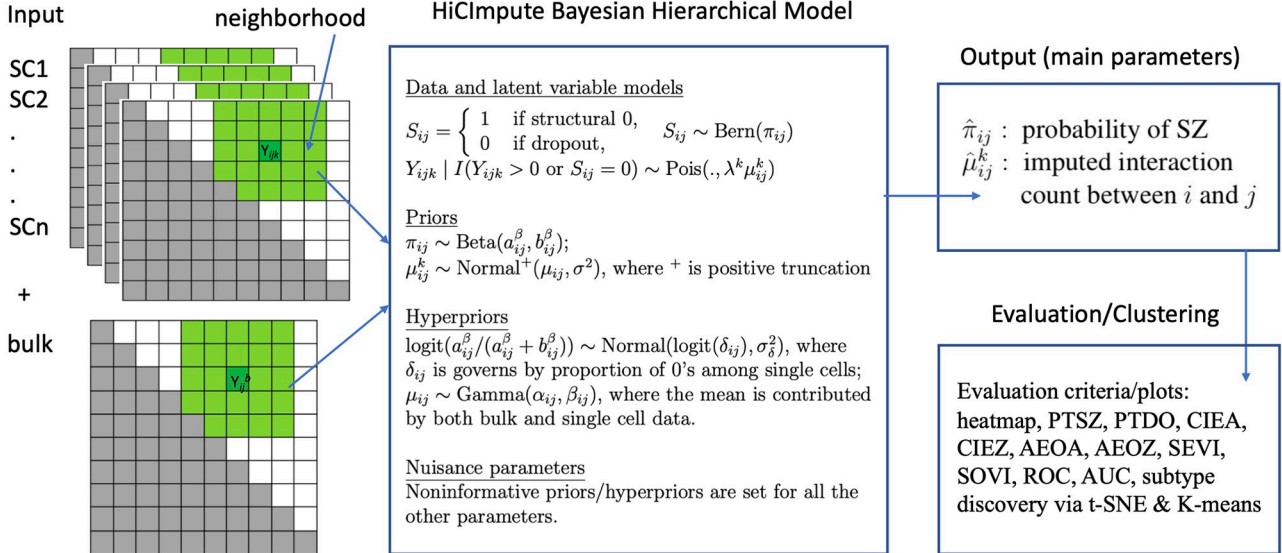

**Fig 1. Schematic of the HiCImpute algorithm.** Each green region on the left denotes the neighborhood. The indicator variable $S_{ij}$ denotes whether an observed zero at the $(i, j)$ position is a structural zero or not. The $\lambda^k$ is related to the sequencing depth of single cell $k$ and acts as a normalizing factor. Finally, the intensity parameter $\mu_{ij}^k$ is assumed to follow a common distribution across all similar single cells; shrinkage estimation with information from neighborhoods and bulk data will be obtained, which provides accommodations for potential overdispersion. PTSZ (proportion of true structural zeros correctly identified), PTDO (proportion of true dropouts correctly identified), AEOA (absolute error of all observed), AEOZ (absolute error of observed zeros), CIEA (correlation between imputed and expected for all observed), CIEZ (correlation between imputed and expected for observed zeros), SEVI (scatterplot of expected versus imputed), and SOVI (scatterplot of observed versus imputed), ROC (receiver operational characteristics), AUC (area under the curve).

correlation (CIEZ and CIEA) and absolute errors (AEOA and AEOZ) for comparing between imputed values and underlying true values, and graphical tools (heatmap, ROC and AUC, SEVI and SOVI) for visualization of imputation accuracy. As part of the workflow, visualization of clustering and subtype discovery results will also be provided via t-SNE and K-means.

## HiCImpute greatly improves data quality

A major goal of imputing scHi-C data is to improve data quality for downstream analyses, including determination of cell identify, clustering, and subtype discovery [14–17]. In addition to HiCImpute, three existing methods that have been used to improve Hi-C data quality are also considered, so that their performance can be investigated and compared to HiCImpute: 2D mean filter (2DMF) in HiCRep [15], 2D Gaussian kernel (2DGK) in SCL [16], and random walk with 3 steps (RW3S) in GenomeDISCO [17]. These three particular methods were selected for comparison because of their well-characterized and known features in the statistics literature (2DMF and 2DGK) or because of their frequent use in this particular type of applications (RW3S).

We first simulated three "types" (T1, T2, T3) of single cells Hi-C data modeled after three K562 single cells data publicly available [22]. In addition to considering three cell types, a number of other parameters are also considered for a thorough investigation, including sequencing depth (7K, 4K, 2K) and the number of cells (10, 50, 100). Details of the simulation procedure is described in Methods; here we simply note that the simulation procedure does not follow any of the analysis methods being compared (2DMF, 2DGK, RW3S, and HiCImpute), and thus the evaluation is objective. We first use heatmaps to visualize a 2D data matrix before and after the data quality improvement for each of the methods considered. It is clearly seen that for a T1 single cell at the 4K sequencing depth, HiCImpute was able to denoise and recover the underlying structure well (Fig 2a). On the other hand, whereas 2DMF and 2DGK over-smoothed the image (the main domain structures are still visible, though), RW3S completely lost the domain structure. The superior performance of HiCImpute can also be seen from the scatterplots of the expected versus the imputed (SEVI plots), where the imputed values are highly correlated with the expected, as the point cloud is distributed tightly around a straight line, including the observed zeros (Fig 2b). On the other hand, all three of the comparison methods have point clouds that follow a funnel shape, indicating much greater variability for larger counts; that is, the imputation becomes less accurate for larger counts. The shrinkage effect is expected (i.e. the imputed values are smaller than the expected counts due to smoothing), although the effect is much more pronounced with the comparison methods than with HiCImpute. Considering the aggregate performance for all single cells, we see that HiCImpute achieves better correlation between the imputed and expected counts, either for all observed values (CIEA) or only the observed zeros (CIEZ) compared to the other methods (Fig 2c). The absolute error for the observed zeros (AEOZ) or for all observed (AEOA) are much smaller compared to the other methods. The above observation for cell type T1 with sequencing depth at 4K holds to a large extend across cell types and number of cells (S1–S4 Figs), although absolute errors for HiCImpute can be slightly larger than the comparison methods for setting with (low) sequencing depth, at 2K.

## HiCImpute is highly sensitive for identifying structural zeros

A novel concept being explored in this paper for scHi-C is structural zeros and our ability to separate them from dropouts. The results discussed thus far (Fig 2b and 2c) provides some indirect assessment of the capability of HiCImpute; we now further provide direct evaluation and comparison with other methods. The results using the Bayes rule (Methods) show that

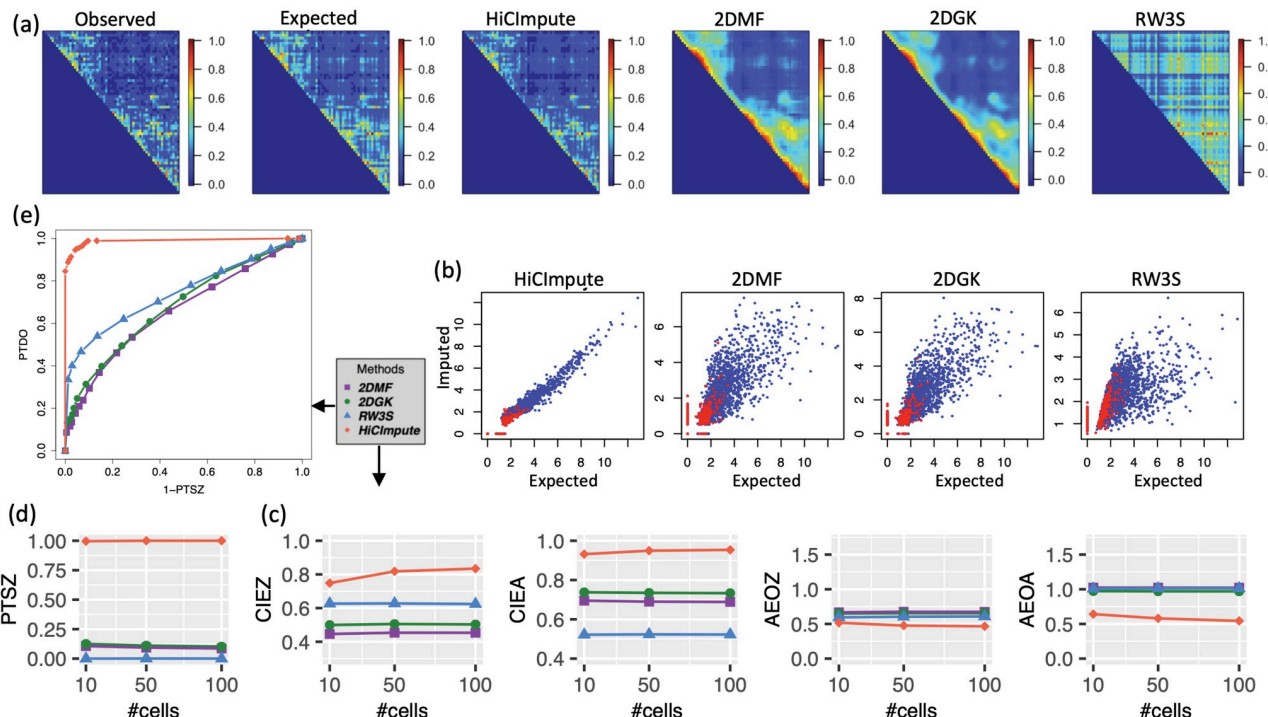

**Fig 2. Comparison of results from HiCImpute for data quality improvement with 2DMF, 2DGK, and RW3S for T1 cells at 4K sequencing depth.**
Ordering of the subfigures is clockwise. (a) Heatmaps of the first single cell showing the observed and true (expected) 2D matrix images as well as the
results from HiCImpute and three comparison methods; (b) Scatterplots of Extected Versus Imputed (SEVI plots) for HiCImpute and the comparison
methods—the red dots represent the observed zeros, which contain both true SZs (expected = 0) and DOs; (c) aggregate results (over single cells) based
on several evaluation criteria; (d) Proportion of true SZs correctly detection averaged over single cells; (e) ROC curves accounting for both PTSZ and
PTDO (with AUC = 0.98 compared to 0.66, 0.68, and 0.74 for 2DMF, 2DGK, and RW3S, respectively).

HiCImpute has an extremely high sensitivity for detecting SZs. In fact, using the criterion of
the proportion of true structural zeros (PTSZ) detected (i.e. the proportion of true underlying
structural zeros being correctly declared as SZs—sensitivity), HiCImpute reaches the propor-
tion of greater than 0.95 for all the situations considered (Fig 2d and S3 Fig). For the three
comparison methods, an observed zero is identified as SZ if its imputed value, in the original
scale, is less than 0.5. This criterion, borrowed from the existing literature on scRNA-seq [7,
11], led to subpar performances: for T1 4K, less than 0.25 of the true SZs were detected.
Although the PTSZ may reach over 0.75 when the sequencing depth is low (e.g. T2 2K), the
value is typically low, at about 0.25 or less for most of the settings considered.

Since the three comparison methods only aim for data quality improvement, not for identi-
fying SZs, their adaptation for this purpose with the threshold of 0.5 may be viewed as arbitrary.
Therefore, we explore a range of threshold values and plot the performance as ROC curves (Fig
2e and S4 Fig). Once again, HiCImpute outperforms the other methods for this evaluation crite-
rion. HiCImpute not only has larger sensitivity for detecting SZs, but also larger specificity for
detecting DOs, with a much larger area under the curve (AUC). For HiCImpute, the AUC is
0.98 compared to 0.66, 0.68, and 0.74 for 2DMF, 2DGK, and RW3S, respectively.

## HiCImpute identifies DOs and imputes them with high accuracy

Fixing the PTSZ at 0.95, we further examined and compared the performances of the methods.
The reason that we chose to fix the threshold at this level is akin to controlling for the type II

**Table 1. Mean (standard error) of the proportion of true dropouts (PTDO) correctly detected when the detection rate for the proportion of true structural zeros (PTSZ) is set to be 0.95.**

| Type | Sequence depth | #cells | HiCImpute | 2DMF | 2DGK | RW3S |
|------|----------------|--------|-----------|------|------|------|
| T1 | 7k | 10 | 0.98 (0.01) | 0.29 (0.04) | 0.31 (0.04) | 0.50 (0.06) |
|  |  | 50 | 0.99 (0.01) | 0.27 (0.05) | 0.31 (0.05) | 0.47 (0.07) |
|  |  | 100 | 0.99 (0.01) | 0.27 (0.04) | 0.30 (0.05) | 0.46 (0.06) |
|  | 4k | 10 | 0.95 (0.01) | 0.21 (0.03) | 0.24 (0.03) | 0.43 (0.03) |
|  |  | 50 | 0.95 (0.01) | 0.18 (0.03) | 0.25 (0.03) | 0.44 (0.03) |
|  |  | 100 | 0.95 (0.01) | 0.19 (0.03) | 0.26 (0.03) | 0.44 (0.03) |
|  | 2k | 10 | 0.95 (0.00) | 0.39 (0.02) | 0.45 (0.02) | 0.55 (0.02) |
|  |  | 50 | 0.98 (0.00) | 0.39 (0.02) | 0.45 (0.02) | 0.56 (0.03) |
|  |  | 100 | 0.98 (0.00) | 0.39 (0.02) | 0.44 (0.02) | 0.56 (0.03) |
| T2 | 7k | 10 | 0.60 (0.03) | 0.08 (0.03) | 0.10 (0.04) | 0.26 (0.06) |
|  |  | 50 | 0.64 (0.04) | 0.10 (0.04) | 0.11 (0.04) | 0.25 (0.05) |
|  |  | 100 | 0.63 (0.04) | 0.10 (0.03) | 0.11 (0.03) | 0.26 (0.05) |
|  | 4k | 10 | 0.89 (0.01) | 0.30 (0.02) | 0.34 (0.02) | 0.63 (0.03) |
|  |  | 50 | 0.88 (0.01) | 0.29 (0.02) | 0.33 (0.02) | 0.62 (0.03) |
|  |  | 100 | 0.88 (0.01) | 0.29 (0.02) | 0.33 (0.02) | 0.62 (0.03) |
|  | 2k | 10 | 0.93 (0.00) | 0.39 (0.03) | 0.43 (0.03) | 0.76 (0.03) |
|  |  | 50 | 0.95 (0.00) | 0.46 (0.02) | 0.43 (0.02) | 0.76 (0.02) |
|  |  | 100 | 0.96 (0.00) | 0.39 (0.02) | 0.43 (0.02) | 0.76 (0.02) |
| T3 | 7k | 10 | 0.67 (0.02) | 0.07 (0.02) | 0.08 (0.02) | 0.32 (0.04) |
|  |  | 50 | 0.66 (0.03) | 0.07 (0.02) | 0.08 (0.02) | 0.33 (0.05) |
|  |  | 100 | 0.67 (0.03) | 0.06 (0.02) | 0.08 (0.02) | 0.32 (0.05) |
|  | 4k | 10 | 0.91 (0.01) | 0.09 (0.01) | 0.10 (0.01) | 0.54 (0.05) |
|  |  | 50 | 0.89 (0.01) | 0.10 (0.01) | 0.12 (0.01) | 0.53 (0.03) |
|  |  | 100 | 0.89 (0.01) | 0.09 (0.01) | 0.12 (0.02) | 0.54 (0.03) |
|  | 2k | 10 | 0.96 (0.00) | 0.18 (0.02) | 0.19 (0.02) | 0.56 (0.03) |
|  |  | 50 | 0.96 (0.00) | 0.15 (0.01) | 0.19 (0.01) | 0.56 (0.03) |
|  |  | 100 | 0.95 (0.00) | 0.18 (0.01) | 0.19 (0.01) | 0.56 (0.03) |

error at 0.05. Since the ability to identify SZs is critical for downstream analyses such as constructing 3D structures (as a penalty may be imposed based on SZs [5, 23, 24]), it is desirable to keep the proportion of failure to correctly identify the underlying SZs at a low level (e.g. 0.05). One can see from Table 1 that HiCImpute outperforms the other methods for correctly identifying the true DOs by a large margin across all three single cell types, sequencing depth, and sample sizes. For example, for T1 4K, the specificity, PTDO, for HiCImpute is at 95%; in contrast, even among the best of the three methods, RW3S, at most only 44% of the DOs are correctly identified. In general, the specificity for HiCImpute is more than doubling that for a comparison method when the specificity for the method is below 50%. The accuracy of the imputed values for the DOs and the far superior performance of HiCImpute over the three smoothing methods are consistent with the plots discussed earlier (Fig 2e and S4 Fig). These results, presented as ROC curves, indeed show that HiCImpute balances the goal of detecting SZs with that of accurate imputation of DOs.

We further substantiated the above findings in two directions. First, we use two other neighborhoods to study any potential effects due to a particular specification of a neighborhood. The results (S2 Table) show that the outcomes are rather insensitive to the choice, with all three choices giving consistent results. Then, we carried out another analysis using

additional data to gauge whether our results are robust to the choice of underlying cell type structures. Specifically, we replaced T3 with simulated data based on a GM cell (GSM3271347) [25] while keeping T1 and T2 the same. The analysis results (S3 Table) lead to the same conclusion: HiCImpute performs well and better than the other methods across the different simulated cell types. These results are not surprising: although the original data were simulated based on three K562 cells, their profiles of structural zeros and sampling zeros were made different to simulate "subtype differences." Taken together, these results point to the robust performance of HiCImpute.

## Improved data lead to more accurate clustering of cells

We consider three real scHi-C datasets to demonstrate the improvement of cell type clustering after data improvement with HiCImpute and compare with the results using data improved by the three comparison methods: 2DMF, 2DGK, and WR3S.

The first scHi-C dataset (https://www.ncbi.nlm.nih.gov/geo/query/acc.cgi?acc=GSE117874) consists of 14 GM (lymphoblastoid) and 18 PBMC (peripheral blood mononuclear cells) [25]. Based on a sub-2D matrix of dimension $63 \times 63$ on chromosome 20 of the 32 SCs of the observed Hi-C data and using the K-means algorithm, there was one misclassification for the GM and 7 for PBMC (Table 2a). With the imputed data from HiCImpute and the same sub-2D matrix, all cells were correctly classified. On the other hand, using imputed data by 2DMF and 2DGK do not see any improvement, whereas the WR3S imputed data in fact led to more misclassifications on the GM and PBMC cells than using the observed data. The scatterplot of observed versus imputed (SOVI plot) shows that the imputed data from HiCImpute are highly correlated with the observed, whereas the other methods see widely scattered point clouds (S5 Fig). The correlation between the observed and the imputed are also seen to be much higher across all cells (S6 Fig).

The second Hi-C dataset (https://www.ncbi.nlm.nih.gov/geo/query/acc.cgi?acc=GSE80006) consists of two bulk K562 Hi-C data—one K562A (bulk A) and one K562B (bulk B)—and 19 scHi-C data of K562A and 15 K562B cells [22]. However, among the 34 single cells, only 10 has sequencing depth over 5K; for the remaining ones, most only have sequencing depth of 1K. Using hierarchical clustering, one can see that K562A and K562B cells are mixed together, and in fact, the group in the middle consists of the 10 cells that have sequencing depth of at least 5000, together with the two bulk data (S7 Fig). Considering only these 10 singles cells and clustering them using K-means based on the observed data led to one of the two K562A cells clustered with the eight K562B cells (Table 2b). On the other hand, clustering

**Table 2. Clustering results for three single cell Hi-C datasets before and after data improved with four methods.**

| (a) GSE117874 | | Observed | | HiCImpute | | 2DMF | | 2DGK | | RW3S | |
|---|---|---|---|---|---|---|---|---|---|---|---|
| | | C1 | C2 | C1 | C2 | C1 | C2 | C1 | C2 | C1 | C2 |
| | GM | 13 | 1 | 14 | 0 | 13 | 1 | 13 | 1 | 11 | 3 |
| | PBMC | 7 | 11 | 3 | 15 | 7 | 11 | 7 | 11 | 8 | 10 |
| (b) GSE80006 | | Observed | | HiCImpute | | 2DMF | | 2DGK | | RW3S | |
| | | C1 | C2 | C1 | C2 | C1 | C2 | C1 | C2 | C1 | C2 |
| | K562A | 1 | 1 | 2 | 0 | 1 | 1 | 1 | 1 | 1 | 1 |
| | K562B | 0 | 8 | 0 | 8 | 0 | 8 | 0 | 8 | 0 | 8 |
| (c) scm3C-seq | | Observed | | HiCImpute | | 2DMF | | 2DGK | | RW3S | |
| | | C1 | C2 | C1 | C2 | C1 | C2 | C1 | C2 | C1 | C2 |
| | L4 | 76 | 55 | 131 | 0 | 77 | 54 | 77 | 54 | 76 | 55 |
| | L5 | 105 | 75 | 0 | 180 | 105 | 75 | 104 | 76 | 105 | 75 |

using improved data from HiCImpute corrected the misclassification, resulting in perfect separation of the K562A and K562B cells. In contrast, using data improved by 2DMF, 2DGK, or RW3S did not yield any improvement over the outcome from simply using the observed data. SOVI plots and correlations between observed and imputed further substantiate the superior performance of HiCImpute (S5 and S6 Figs).

The third scHi-C dataset (https://github.com/dixonlab/scm3C-seq) consists of prefrontal cortex cells of subtypes L4 (131 cells) and L5 (180 cells) [26]. It is known that there are 14 cell subtypes of the prefrontal cortex cells, including eight neuronal subtypes that were all clustered together based on the observed scHi-C data [26]. Among them are L4 and L5, two excitatory neuronal subtypes known to be located on different cortical layers. Our K-means analysis based on the observed L4 and L5 scHi-C data shows that these two subtypes are indeed mixed together (Table 2c), echoing the earlier finding [26]. Although the problem is much more challenging compared to the first two datasets given its size and the extremely mixed clustering results based on the observed data, using data improved by HiCImpute led to perfect separation of the two subtypes; whereas none of the data improved using the comparison methods yielded any improvement. SOVI plots of the observed versus the imputed values and the correlations across all cells painted the same picture as for the other two datasets on the superiority of scHi-C over the other methods (S5 and S6 Figs). It is reassuring to see that, for this much larger dataset (compared to the first two), there continues to be good agreement between the imputed and observed non-zeros, pointing to the accuracy of improvement for the inference on the observed zeros.

## Discovery of subtypes of L4 and L5

Cell to cell variability is a driving force behind the developments of single cell technologies [27]. Based on single cell RNA-seq data, subtypes of L4 and L5 have been discovered. For example, two L4 subtypes, Exc L4–5 FEZF2 SCN4B and Exc L4–6 FEZF2 IL26, were found to be highly distinctive as they occupied separate branches of a dendrogram [28]. On the other hand, the L4–IT–VISp–Rspo1 cells were shown to exhibit heterogeneity along the first principal component of scRNA-seq data [29]. Similarly, two subtypes of L5, Exc L5–6 THEMIS C1QL3 and L5–6 THEMIS DCSTAMP, were also found to be on two separate branches of a dendrogram [28], while there was also research that further classified L5 cells into L5a and L5b subtypes [29]. Other works have also found subclusters of excitatory neurons including L4 and L5 [30–32].

Inspired by the ample evidence in the literature that subtypes of L4 and L5 exist, we visualized the observed data and those improved by HiCImpute, 2DMF, 2DGK, and RW3S using t-SNE and then clustered using K-means. Based on the within-cluster sum of squares and visually inspecting the number of clusters where the "elbow" is identified (Fig 3a), we see that there are two clusters for the observed data (Fig 3b) and those improved with 2DMF, 2DGK, or RW3S (Fig 3d). On the other hand, for the data improved with HiCImpute, the plot clearly shows the existence of four clusters (Fig 3c). In fact, these four clusters are very well separated, with two of them consisting of purely L4 cells and two L5 cells. Using the adjusted rand index (ARI) [33], we further investigate the optimal number of clusters and the performance of clustering for the observed data and improved data with HiCImpute and the other methods. Based on the results (S4 Table), it is without a doubt that HiCImpute improves over the observed data and outperform the other methods. Most importantly, using data improved with HiCImpute, two subtypes each for L4 and L5 emerge, consistent with results in the literature. On the other hand, none of the data improved with the other methods led to the discovery of any subtypes for L4 or L5.

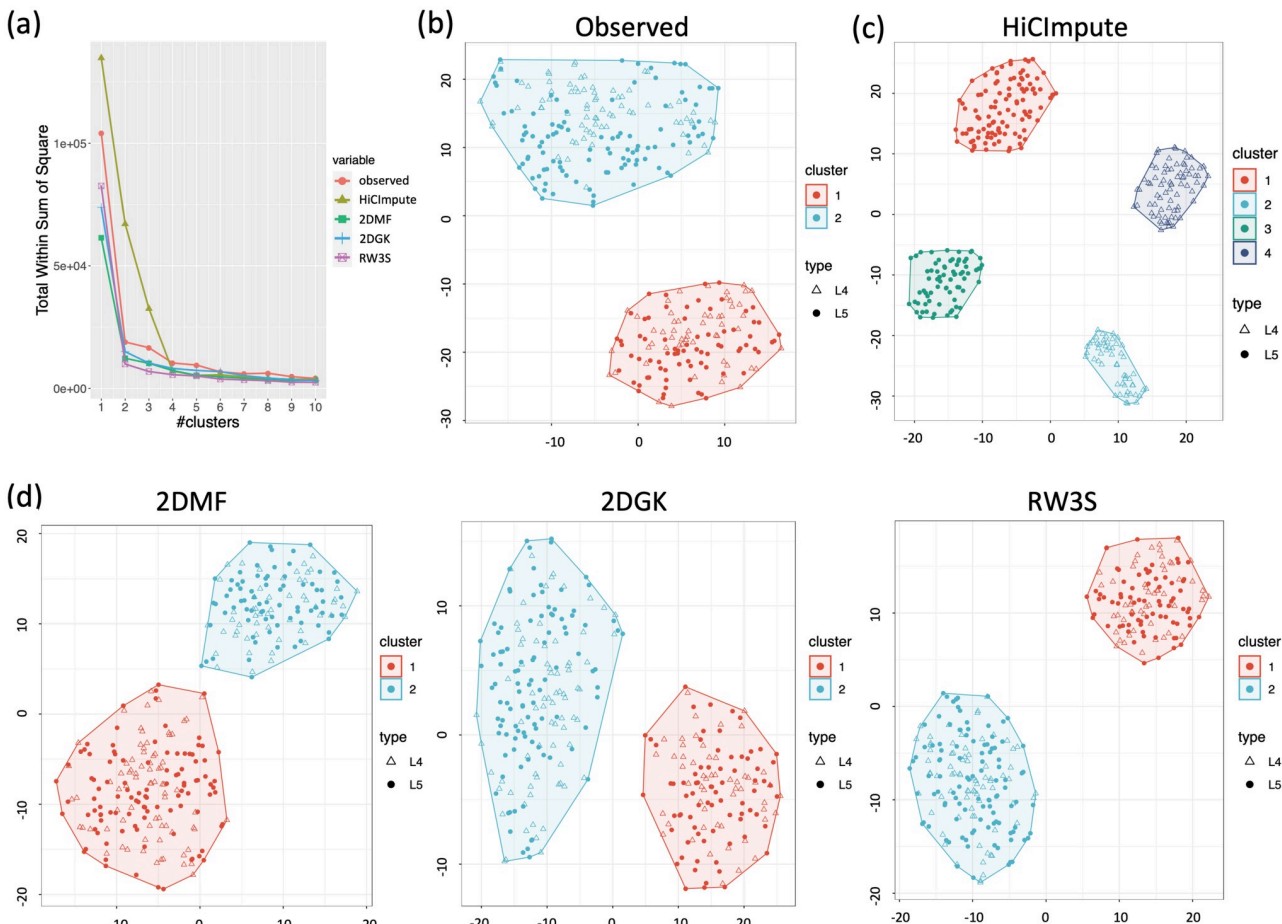

**Fig 3. Comparison of results from HiCImpute, 2DMF, 2DGK, and RW3S via t-SNE visualization and clustering with K-means.** (a) Plots of total within-cluster sum of squares versus number of clusters for K-means analysis; (b) t-SNE visualiation and K-means clustering boundaries based on observed data; (c) Same as (b) but based on HiCImpute-improved data; (d) Same as (b) but based on 2DMF, 2DGK, or RW3S-improved data.

Visualization by a 2-way clustering heatmap using normalized and log-transformed HiCImpute-improved data for the 500 positions (on the 2D matrix) with the highest variation across all cells further substantiates the 3D structural differences between each of the two subtypes of L4 and L5 (Fig 4a), where the L4 cells were clustered into two subgroups, and the same for the L5 cells. Several genes that were found to be differentially expressed among subgroups of L4 and L5 in the literature [28] were also marked on the heatmap, where it can be seen that there are differential interaction intensities among the subtypes. To further elucidate the potential correspondence between differential gene expression and differential 3D interaction intensities among the subtypes of L4 or among those of L5, we combined all cells from each of the subtypes into four mega 2D matrices (L4T1, L4T2, L5T1, L5T2) and normalized them to the same total count and scaled them to be a value between 0 and 1. These 2D matrices displayed as heatmaps exhibit regions having differential interaction intensities. Zooming in on the region chr20:35,000,000–55,000,000, we can see that L5T1 has relatively lower intensities compared to its L5T2 counterparts, and furthermore, the latter appears to have some subtle domain structures that are missing in the former (Fig 4b). Interestingly, when we reproduce the mean RNAseq data in the same region for two subtypes discussed in the

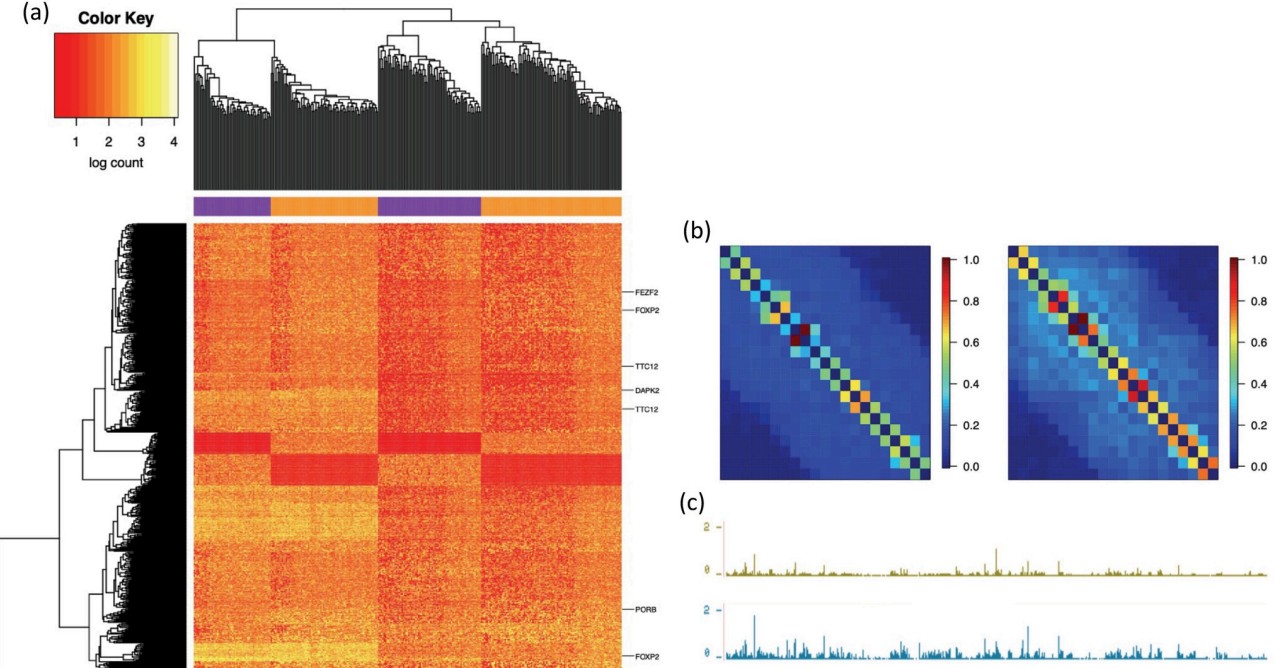

**Fig 4. Correspondence between differential 3D structures and differential gene expressions among further subtypes.** (a) Heatmap of 500 positions in the 2D interaction matrix with the largest variation among the cells (each row is a position and each column is a cell), with a 2-way clustering outcome placing the L4 cells into the two purple groups and the L5 cells into the two orange groups, and genes showing differential expression [28] indicated on the right edge of the heatmap; (b) Mega 2D matrices of normalized and scaled interaction intensities displayed as heatmaps, with the left for L5T1 and right for L5T2; (c) Mean gene expression for two L5 subtypes described in the literature [28].

literature [28] as tracks in the UCSC genome browser (https://human-mtg-rna-hub.s3-us-west-2.amazonaws.com/HumanMTGRNAHub.html), the differences in the gene expression patterns are obvious (Fig 4c). Examples of other regions where there appear to be a correspondence between 3D structure differences and gene expression differences among subtypes of L4 or L5 are also provided (S11 Fig).

## Discussion

This paper introduces the concept of structural zeros in the context of 3D contacts, and explores the ability of HiCImpute for separating structural zeros from dropouts and the accuracy of imputing the dropout values. From both simulation and real data studies, we can see that HiCImpute has great ability of identifying structural zeros, and outperforms existing methods for its accuracy of imputing the contact counts of dropouts based on multiple criteria. This conclusion is based on outcomes from considering a number of factors, including the number of cells, sequencing depth, multiple cell types, and whether bulk data are available. The improved data from HiCImpute has greatly impacted downstream analysis. From the examples of clustering GM and PBMC cells, K562 cells, and prefrontal cortex cells, we have seen that data improved with HiCImpute led to more accurate clustering judging from known cell types. What is most exciting is the ability of HiCImpute for producing improved data that can lead to not only the separation of L4 and L5 of the prefrontal cortex cells, but also the discovery of two subtypes, each within L4 and L5, for the first time using scHi-C data. Given that the existence of further subtypes within each of these two excitatory neuronal subtypes has

been documented in the literature using scRNA-seq data [28–32, 34], and given that our own analysis has found regions where there are differential expression and differential interactions between further subtypes, our results may be viewed as evidence for the potential of establishing the correspondence between scHi-C and scRNA and elucidating the ability of scHi-C data, when appropriately enhanced, for investigating cell-to-cell variability and uncovering hidden subpopulations and substructures.

The Bayesian hierarchical modeling framework affords us the flexibility of integrating multiple pieces of additional information, including bulk and other single cell data. In particular, through multi-level modeling, we can represent the interplay between bulk and scHi-C data on their influence exerted upon the overall mean interaction counts. Although heterogeneity is likely to be present in both bulk and single cell data, the integration of their information on a localized scale (the neighborhoods) and the much more accurate observed counts in bulk data are likely to help with delineating structural zeros and dropouts, and help with accurate imputation of the dropout values. Results from our simulation and real data analysis, including the biological relevance of downstream analysis from the HiCImpute-improved data, corroborate the appropriateness of the modeling scheme and the inference procedure.

As the goal of this work is on improving data quality through identifying structural zeros and accurately imputing dropouts, we assume known major cell types. Given the potential heterogeneity even within the same cell type, subtypes may exist, and the quality-improved data may be able to aid in detecting these subtypes. As demonstrated in our three data applications, the cell type information in each case was known. We further demonstrated that, with the prefrontal cortex cells of types L4 and L5, we were indeed able to find subtypes within each of them, consistent with the literature using single-cell RNA-seq data.

The more accurate results do not come without a greater cost in computational time, though. Our scHi-C method is implemented in C++ for computational efficiency since the algorithm based on Markov chain Monte Carlo is computationally intensive. The computational time for HiCImpute was in hours with hundred of cells for the L4/L5 prefrontal cortex data, compared to minutes with the other methods (S5 Table). Considering the time needed for collecting the samples and generating the data, this price to pay is completely justifiable, especially since biological insights are gained with the improved data from HiCImpute compared to the alternatives in the literature. Hours of computational time for the "truth" to be revealed is certainly worth the wait and the cost compared to the "truth" continued to be hidden. Nevertheless, effort will continue to be made to further improve the computational efficiency.

## Materials and methods

### Bayesian hierarchical model

Suppose we have contact matrices for $K$ Single Cells (SCs) and a bulk Hi-C dataset that is related to these SCs. Let $Y_{ijk}$ and $Y_{ij}^b$ represent the observed interaction frequencies between loci $i$ and $j(i < j)$ for SC $k$, $k = 1, \cdots, K$, and the bulk data, respectively. Among those observed zeros, some are true zeros (i.e. structural zeros, SZs) since the two loci never interact with each other in this particular cell; whereas others are sampling zeros (i,e. dropouts, DOs) since they interact infrequently and thus dropout from the sample as their interaction is not observed due to insufficient sequencing depth. This zero-inflated problem is complicated since not all zeros are created equal, and our goal is to make statistical inferences to tease out those that are SZs from those that are DOs, and to impoute the values for the DOs.

Since $Y_{ijk}$ a count, its distribution can be reasonably modeled by a Poisson distribution, with additional hierarchical modeling to address potential overdispersion, leading to

equivalency with a negative binomial model. Let $T_k = \Sigma_{i<j} Y_{ijk}$ denote the sequencing depth of SC $k$, and let $\mu_{ij}^k$ be the parameter representing the intensity of SC $k$ if the SC is depth-normalized to a desired sequencing depth $T$, which may be the maximum sequencing depth among the SCs, that is, $T = \max\{T_k, k = 1, \cdots, K\}$, or may simply be an intended sequencing-depth level appropriate for downstream analysis, say 300,000, the level of the best K562 scHi-C data [22]. Then $\lambda^k = T_k/T$ is the proportionate sequencing depth of SC $k$ relative to the intended one.

To distinguish the SZs from the DOs, we define an indicator variable $S_{ij}$, which equals to 1 if loci $i$ and $j$ do not interact, otherwise it is 0. That is, $S_{ij} \sim \text{Bernoulli}(\pi_{ij})$, where $\pi_{ij}$ is the probability that pair $i$ and $j$ do not interact. $Y_{ijk}$ therefore follows a mixture of a point-mass distribution at 0 and a Poisson distribution with mixing proportions $\pi_{ij}$ and $1 - \pi_{ij}$, respectively. Hence, $Y_{ijk} \mid I(Y_{ijk} > 0 \text{ or } S_{ij} = 0) \sim \text{Poisson}(\lambda^k \mu_{ij}^k)$, where $I(\cdot)$ is the usual indicator function, and $\lambda^k \mu_{ij}^k$ is the intensity parameter for the non-normalized observed counts. We further let $\pi_{ij}$ follow a Beta distribution and its mean is governed by the observed proportion of zeros across the SCs in that position. The idea is that if there is a large proportion of zeros at that position, it is more likely to be an SZ.

We allow for cell-to-cell variability by setting up an additional hierarchy to model $\mu_{ij}^k$ as follows: $\mu_{ij}^k \sim \text{Normal}^+(\mu_{ij}, \sigma_{ij}^2)$, where $\text{Normal}^+$ is a truncated normal distribution on positive numbers, $\sigma_{ij}^2$ is taken to be the standard deviation of nonzero counts in a neighborhood centered at $(i, j)$, and $\mu_{ij}$ is further assumed to follow a Gamma distribution whose mean borrows information from both the bulk Hi-C and the neighborhood data across similar SCs. Specifically, let $Y_{ij}^{(nSC)} = \sum_k Y_{ijk}/\sum_k \lambda_k$, which is the weighted average of the "normalized" (to sequencing depth $T$) contacts between $i$ and $j$ over the SCs with the weight proportional to the sequencing depth of each SC. Similarly, we let $T^b = \sum_{i<j} Y_{ij}^b$ and $Y_{ij}^{(nB)} = TY_{ij}^b/T^b$ be the sequencing depth of the bulk data and the count of the bulk data "normalized" to sequencing depth $T$, respectively. Then the mean of the Gamma distribution is set to be $(\sum_{(i,j)\in\Omega_2} Y_{ij}^{(nB)}/\|\Omega_2\|)(\sum_{(i,j)\in\Omega_1} Y_{ij}^{(nSC)}/(\|\Omega_1\|\bar{Y}^{(nSC)}))$, where $\Omega_1, \Omega_2$ are the neighborhoods for the SCs and the bulk data, respectively, $\|\cdot\|$ is the cardinality of the neighborhood, and $\bar{Y}^{(nSC)}$ is the average of the $Y_{ij}^{(nSC)}$ over the SCs. Under this setting, we first use the normalized neighborhood-mean of the bulk data to set the initial value, as such data are likely to be more reliable given its much larger sequencing depth. Nevertheless, information from the SCs plays a modifying, but crucial, role by providing a weight factor for the initial value: if the average count in the neighborhood of the SCs is larger than the average count over the entire matrix, then the mean neighborhood count of the bulk data will be boosted; otherwise it will be shrunk. Thus, the information from bulk and single cells are intertwined in their collective effect on the overall mean counts. Throughout all the data analysis, the neighborhood is taken to be the two immediate neighbors (if available) in all directions of a lattice (Fig 1).

Details on the prior specifications and the full posterior distribution are provided in S1 Text. Due to the fact that the posterior distribution is known only up to a constant of proportionality, we are unable to make inference on the parameters from the posterior distribution analytically. Therefore, we opted to use a Markov chain Monte Carlo (MCMC) sampling procedure, with the detailed updating schemes and convergence diagnostics also provided in S1 Text. Using samples generated by MCMC from the posterior distribution of $\pi$ for a particular pair that have an observed zero count in an SC, we can make inference about whether the zero is a SZ or a DO. A natural decision based on the Bayes rule is to declare a zero for an SC to be a SZ if the corresponding $\pi$ is estimated by the posterior sample mean to be greater than 0.5.

However, to compare between HiCImpute and existing methods, as described in more details in the evaluation criteria below, we also set different thresholds to obtain an ROC curve.

For comparison with 2DMF, 2DGK, and RW3S in terms of PTSZ, we follow the recommendation in the scRNA-seq literature by labelling an observed zero to be a structural zero if the imputed count is less than 0.5 for each of the comparison methods [11]. We also vary the threshold to obtain an ROC curve separately for each of the three methods.

## Performance evaluation criteria

To evaluate the performance of HiCImpute and to compare with other data quality improvement methods, including 2DMF, 2DGK, and RW3S, we consider the following novel criteria in addition to standard measures and plots, including the heatmap and t-SNE visualization tools, receiver operating characteristic (ROC) curves and area under the curve (AUC), K-means clustering algorithm, and the adjusted rand index for evaluating clustering results.

- PTSZ: Proportion of true structural zeros correctly identified. This is defined as the proportion of underlying structural zeros that are correctly identified as such by a method. Being able to separate structural zeros from dropouts is important for downstream analyses, especially for single cell classification to reveal cell sub-populations.

- PTDO: Proportion of true dropouts correctly identified. This is defined as the proportion of underlying dropouts (due to insufficient sequencing depth) that are correctly identified as such by a method. Similarly, being able to correctly identify dropouts is also critical for a number of downstream analyses.

- SEVI: Scatterplot of expected versus imputed. This serves as a visualization tool to directly assess whether dropouts are correctly recovered and accurately imputed for simulated data where the ground truth is known.

- SOVI: Scatterplot of observed versus imputed—applicable to real data for non-zero observed counts. This serves as a visualization tool to *indirectly* assess whether the imputed values are sensible for the observed zeros by looking at the performance for observed non-zeros. For real data, whether an observed zero is a SZ or DO is unknown, and if it is a DO, the underlying expected non-zero value is also unknown. Nevertheless, the imputed values for the non-zero observed counts should not deviate wildly from the observed values even though some level of "smoothing" is applied.

- CIEA: Correlation between imputed and expected for all observed. This provides a summary statistic to assess how well the imputed values are correlated with their underlying expected counterparts.

- CIEZ: Correlation between imputed and expected for observed zeros. This also provides a summary statistic to assess how well the imputed values are correlated with their underlying expected counterparts, although the focus is only on observed zeros, as they are the main concern.

- AEOA: Absolute errors for all observed data. This is defined as the absolute difference between the imputed and the expected for all observed data. This measure is to gage how well the imputed values can approximate its underlying true values.

- AEOZ: Absolute errors for observed zeros. Unlike AEOA that considers all observed, this measure only considers observed zeros. This measure provides a more focused evaluation on correct identification of structural zeros and the accuracy of the imputing dropout values.

## Simulation studies and settings

To evaluate HiCImpute and compared with the three data quality improvement methods in the literature, we carried out an extensive simulation study for a total of over one hundred settings, including three types of single cells (T1, T2, T3, mimicking three K562 cells [22]), three sequencing depth in a $61 \times 61$ contact matrix on a segment of chromosome 19 (7k, 4k, and 2k), 3 sample sizes (10, 50, 100, representing the number of single cells), and 4 settings of SZs and DOs. Note that the three sequencing depth considered are comparable with the best-quality scHi-C data because we only consider a submatrix of size 61 following an earlier study [14]. The following describes the detailed simulation procedure to generate single cell data for each of the settings as well as bulk data.

- Step 1. Calculate the 3D distance matrix $d$ where $d_{ij} = \sqrt{(x_i - x_j)^2 + (y_i - y_j)^2 + (z_i - z_j)^2}$ at each pair of loci $(i, j)$, $i < j$, where $(x_i, y_i, z_i)$ represents the 3D coordinates for locus $i$ of the 3D structure. For each of the three cells, its 3D structure was constructed using SIMBA3D [5] based on a K562 scHi-C data. [22].

- Step 2. Use the following formula to generate the $\lambda$ matrix following the literature [35]:

$$\log(\lambda_{ij}) = \alpha_0 + \alpha_1 \log d_{ij} + \beta_l \log(x_{l,i}x_{l,j}) + \beta_g \log(x_{g,i}x_{g,j}) + \beta_m log(x_{m,i}x_{m,j}),$$

  where $\alpha_1$ is set to -1 to follow the typical biophysical model, $\alpha_0$ is the scale parameter, and set to be 5.7, 6.3, and 6.8 for the three cell types, respectively. On the other hand, $x_{l,i}$, $x_{g,i}$, and $x_{m,i}$ are covariates generated from uniform distributions to mimic fragment length, GC content, and mappability score, respectively, and their coefficients, the $\beta$'s, are all set to be 0.9.

- Step 3. Find the lower $\gamma$% quantile of the $\lambda_{ij}$ as the threshold, for those $\lambda_{ij} <$ threshold, randomly select half of them to be candidates for structural zeros. Among these candidates, randomly select $\eta$% of them and set their new $\lambda_{ij}$ value to be zero. These are the SZs across all SCs. In our simulation, we consider $\gamma = 10\%$, 20% and $\eta = 80\%$, 50%, leading to 4 combinations. In the results presented in this paper, we only show those for $\gamma = 10\%$ and $\eta = 80\%$. Note that the results for the other three combinations led to the same conclusions qualitatively.

- Step 4. For the remaining $(1 - \eta\%)$, they are randomly set to be SZ or not with equal probabilities when we simulate the contact count matrix for each single cell. For a particular single cell, the new $\lambda_{ij}$ value is set to be zero if a position is selected to be SZ; otherwise, the $\lambda_{ij}$ value is left unchanged in the original $\lambda$ matrix. This leads to be a $\lambda^*$ matrix for a specific single cell. Therefore, the SZs among the $(1 - \eta\%)$ positions vary from SC to SC.

- Step 5. Simulate a 2D contact matrix for a SC using the $\lambda^*$ matrix; the contact count at each position is generated based on a Poisson distribution with the corresponding value in the $\lambda^*$ matrix as the intensity parameter. Note that the count is set to zero (SZ) if the corresponding value in the $\lambda^*$ matrix is zero. Also note that a zero may still result even if the corresponding value is not zero, and these are DOs. This completes the simulation of one SC; the SZs and DOs vary from SC to SC.

- Repeat steps 4 and 5 for as many time as needed to obtain the desired number of SCs (sample size). We consider three sample sizes: 10, 50, and 100 SCs.

As expected, the observed data (including both the SZs and DOs) are much sparser than the expected data (only containing the SZs), especially for observed data with lower sequencing

depth (S6 Table). Finally, we created bulk data by combining the 2D contact matrices from 540 SCs equally divided among the three cell types (180 for each type).

## Supporting information

**S1 Text. Prior specifications, full posterior distributions, sampling procedure, and convergence diagnostics.**
(PDF)

**S1 Fig. Heatmap showing the observed and true (expected) 2D matrix images as well as the results from HiCImpute, 2DMF, 2DGK, and RW3S for T1 (a), T2 (b), and T3 (c) cells at 7K (top), 4K (middle) and 2K (bottom) sequencing depth.**
(PDF)

**S2 Fig. Scatterplots of expected versus imputed (SEVI plots) for HiCImpute, 2DMF, 2DGK, and RW3S for T1 (a), T2 (b), and T3 (c) cells at 7K (top), 4K (middle) and 2K (bottom) sequencing depth the red dots represent the observed zeros, which contain both true SZs and DOs.**
(PDF)

**S3 Fig. Aggregate results (over single cells) based on several evaluation criteria for T1 (a), T2 (b), and T3 (c) cells at 7K (top), 4K (middle) and 2K (bottom) sequencing depth.**
(PDF)

**S4 Fig. ROC curves accounting for both specificity and sensitivity of HiCImpute, 2DMF, 2DGK, and RW3S for T1 (row1), T2 (row2), and T3 (row 3) cells at 7K (column1), 4K (column2) and 2K (column 3) sequencing depth.**
(PDF)

**S5 Fig. Scatterplots of observed versus imputed (SOVI plots) from HiCImpute, 2DMF, 2DGK, and RW3S.** (a) GM (top row) and PMBC (bottom row); (b) K562A (top) and K562B (bottom); (c) L4 (top) and L5 (bottom).
(PDF)

**S6 Fig. Boxplot of correlations between the observed and imputed from four methods for three datasets: GSE117874 (left), GSE80006 (middle), and scm3C-seq (right).**
(PDF)

**S7 Fig. Dendrograms of 34 observed K562 single cells Hi-C data and two bulk datasets.** The dendrogram was generated using the "complete" method.
(PDF)

**S8 Fig. Cumulative mean plots of parameters $a$ (a), $\alpha$ (b), $\mu$ (c) and $\pi$ (d) at 3 positions of a dataset with 10 T1 cells at sequence depth 7k.** Recall that $a$ is the shape parameter of the Beta distribution that is the prior of $\pi_{ij}$; $\alpha$ is the shape parameter of Gamma distribution, which is the prior of $\mu_{ij}$; $\mu$ is the mean of $\mu_{ij}^k$; and $\pi$ is the probability that the pair is a structural zero.
(PDF)

**S9 Fig. Trace plots of three chains starting from different points and the density of the parameters in the first chain for several parameters.**
(PDF)

**S10 Fig. Autocorrelation plots for 6 parameters at position $(i, j) = (40, 42)$ for the simulated dataset with 10 T1 cells at 7K sequencing depth: $\mu$ is the overall expectation for all**

single cells, and $\mu^1$ and $\mu^2$ are the realizations in the first and second single cell, respectively.
(PDF)

**S11 Fig. Correspondence between differential 3D structures and differential gene expressions among further subtypes.** (a) L4 subtype1 (left) and subtype2 (right) on chr8:127,000,000-147,000,000, along with the mean RNA-seq on the same region; (b) L4 subtype1 (left) and subtype2 (right) on chr11:105,000,000-125,000,000, along with the mean RNA-seq on the same region; (c) L5 subtype1 (left) and subtype2 (right) on chr18:1,000,000-15,000,000, along with the mean RNA-seq on the same region; (d) L5 subtype1 (left) and subtype2 (right) on chr20:35,000,000-55,000,000, along with the mean RNA-seq on the same region.
(PDF)

**S1 Table. Partial list of existing methods for Hi-C data quality improvement.**
(PDF)

**S2 Table. Mean (standard error) of the proportion of true dropouts (PTDO) correctly detected by HiCImpute when the detection rate for the proportion of true structural zeros (PTSZ) is set to be 0.95–stability of results with different neighborhoods.**
(PDF)

**S3 Table. Mean (standard error) of the proportion of true dropouts (PTDO) correctly detected when the detection rate for the proportion of true structural zeros (PTSZ) is set to be 0.95–robustness to underlying cell-type structures.**
(PDF)

**S4 Table. K-means clustering results of L4 and L5 cells based on t-SNE embbeded data.** We considered 2–6 clusters for HiCImpute-improved data and 2–4 clusters for the rest since the results did not indicate any need for a greater number of clusters.
(PDF)

**S5 Table. Computation time comparison of packages on three real datasets.**
(PDF)

**S6 Table. Sparsity levels (percentages) of observed data and expected data.**
(PDF)

## Acknowledgments

We thank Ms. Yongqi Liu for testing the HiCImpute software package. We would also like to thank the two referees for their constructive comments.

## Author Contributions

**Conceptualization:** Shili Lin.

**Formal analysis:** Qing Xie.

**Funding acquisition:** Victor Jin, Shili Lin.

**Investigation:** Chenggong Han.

**Methodology:** Qing Xie, Shili Lin.

**Software:** Qing Xie.

**Supervision:** Shili Lin.

**Validation:** Qing Xie.

**Visualization:** Qing Xie, Shili Lin.

**Writing – original draft:** Qing Xie, Shili Lin.

**Writing – review & editing:** Qing Xie, Chenggong Han, Victor Jin, Shili Lin.

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
