## [Decision Letter · Decision Letter 0]

11 Jan 2022

Dear Professor Lin,

Thank you very much for submitting your manuscript "HiCImpute: A Bayesian Hierarchical Model for Identifying Structural Zeros and Enhancing Single Cell Hi-C Data" for consideration at PLOS Computational Biology.

As with all papers reviewed by the journal, your manuscript was reviewed by members of the editorial board and by several independent reviewers. In light of the reviews (below this email), we would like to invite the resubmission of a significantly-revised version that takes into account the reviewers' comments.

We have now received detailed reports on your manuscript from 2 referees. You will see from their comments below that, while they find your work of interest, several important points remain that need to be addressed. We would like to consider your response to these concerns in the form of a revised manuscript before we make a final decision on publication.

We cannot make any decision about publication until we have seen the revised manuscript and your response to the reviewers' comments. Your revised manuscript is also likely to be sent to reviewers for further evaluation.

Sincerely,

Eli Zunder

Guest Editor

PLOS Computational Biology

Ilya Ioshikhes

Deputy Editor

PLOS Computational Biology

We have now received detailed reports on your manuscript from 2 referees. You will see from their comments below that, while they find your work of interest, several important points remain that need to be addressed. We would like to consider your response to these concerns in the form of a revised manuscript before we make a final decision on publication.

Reviewer's Responses to Questions

**Comments to the Authors:**

Reviewer #1: Comments to the Author

Xie et al. develop a Bayesian hierarchy model, HiCImpute, to improve single cell Hi-C (scHi-C) data and identify structural and sampling zeros, which focus on addressing an important problem in scHi-C analysis. The article is in general well-written and easy to read. My main concern lies in the insufficient description of HiCImpute. My suggestion is therefore to supplement related content to ensure a clearer description of this model.

Below, I outline my suggestions:

Major suggestions:

1) What is Bayesian Hierarchical Model and why it could be applied to improve scHi-C data and identify structural zeros and sampling zeros. Above question should be described in the introduction.

2) Whether HiCImpute can enhance scHi-C data based on the identification of structural zeros and sampling zeros, that is, whether HiCImpute can only enhance the structural zeros but not sampling zeros.

3) The bulk data used in HiCImpute is combining from single cells? Whether it can apply real bulk Hi-C data and is it necessary to use it. Besides, whether HiCImpute would introduce noise using bulk data as reference to improve Hi-C data. In other words, whether HiCImpute would enhance the signal which exist in bulk data but not in some single cell? In this situation, the heterogeneity of 3D structure may be neglected or incorrectly estimated.

4) Whether HiCImpute could be applied to sample with large numbers of cells? What about the accuracy of improvement and time consumption?

5) Whether the better performance of HiCImpute is related to the simulation procedure carried out in the text?

Minor suggestions:

1) P5: Taking on the challenging problems of separating the zeros into structural zeros and sampling zeros. What’s the challenging problems specifically refers to? And why could Bayesian Hierarchical Model can address these problems?

2) It’s better to adjust Figure 1 to show the schematic of the HiCImpute algorithm more clearly.

3) How to validate the cluster results are more accurate after data improved in Table2, especially in GM and K562 cell line, which are difficult to explain the biological meaning of cluster results? Number of clusters is not a sufficient criterion for judging the accuracy of classification.

4) There are a few format problems here and there. Besides, the abbreviations used in the text should be checked carefully, some words (e.g. structural zeros) are abbreviated repeatedly and the full name of these words appear after abbreviated.

P7L17: ...consider.It.. There should be a space between ‘.’ and It.

P16L17: Among those observed 0’s, some are true 0s (i.e. structural zeros, SZs) ..., irregular description.

5) It’s better to unify the use of proper nouns (e.g. sampling zeros and dropouts) to avoid confusing readers.

6) Please confirm the format requirement of the reference. The style of reference is not consistent with the article published recently.

Reviewer #2: Xie et al. propose a method named HiCImpute to impute zeros in single-cell HiC data, while trying to differentiate between structural zeros and sampling zeros. The authors claim that HiCImpute outperforms three existing methods based on a set of simulated and real data studies. I agree this is an important research topic, but additional analyses or clarifications are in need to improve the validity of the method and strength of the claim. Below are my detailed comments.

The sequencing depth of the best K562 scHi-C data is 300,000. Why did the authors only consider sequencing depths of 2K, 4K, and 7K in the simulation? Can the authors show results on data with larger sequencing depths?

The simulated contact matrix is 61 by 61. How is this dimension selected? This is quite small compared with the actual contact matrices.

What’s the difference between data of T1, T2, and T3? The Method section mentions they are mimicking three K562 cells, but there are essentially the same cell type. It would be more interesting to evaluate the methods using simulated data of different cell types, to test the methods’ performance given different data structures.

From the simulated data, the sparsity levels of the observed data and expected data are very similar (Figure 2). Based on the authors’ description of single cell HiC data in Introduction, the observed data is supposed to be much sparser than expected data.

“For the three comparison methods, an observed zero is identified as SZ if its imputed value is less than 0.5.” Is this threshold applied to count data or normalized data? If normalized data is used, how is the normalization done?

Can the authors clarify if HiCImpute is the only method that requires bulk data among the four compared methods? This is not very clear in the current manuscript.

It seems that the proposed Bayesian hierarchy model requires prior information of cell types or clusters, since the model assumes that the K cells have the same pattern of SZs. Therefore, it is not clear how HiCImpute is used to improve clustering analysis in the real data applications. Was HiCImpute applied to single cell data without using any cell type information? How does the model distinguish different cell types during the imputation process?

For the mean of the Gamma distribution, how is the neighborhood defined? Is this done in a data-adaptive manner? Does this range affect the imputation performance?

Can the authors explain why this mean is defined based on the ratio between bulk and single-cell data?

Does the posterior distribution have a closed-form representation? If yes, this should be added to the main text. It would also be helpful to list the full model (in math notations) after explaining individual model assumptions in Method.

**Have the authors made all data and (if applicable) computational code underlying the findings in their manuscript fully available?**

Reviewer #1: Yes

Reviewer #2: None

PLOS authors have the option to publish the peer review history of their article (what does this mean?). If published, this will include your full peer review and any attached files.

Reviewer #1: **Yes: **Xiaochen Bo

Reviewer #2: No
---

## [Decision Letter · Decision Letter 1]

21 Apr 2022

Dear Professor Lin,

We are pleased to inform you that your manuscript 'HiCImpute: A Bayesian Hierarchical Model for Identifying Structural Zeros and Enhancing Single Cell Hi-C Data' has been provisionally accepted for publication in PLOS Computational Biology.

Best regards,

Eli Zunder

Guest Editor

PLOS Computational Biology

Ilya Ioshikhes

Deputy Editor

PLOS Computational Biology

Dear Dr. Lin and co-authors,

Thank you very much for submitting your revised manuscript files. I think it looks great, and our two reviewers both commented that all their concerns were addressed. I am pleased to inform you that I'm recommending your manuscript, "HiCImpute: A Bayesian Hierarchical Model for Identifying Structural Zeros and Enhancing Single Cell Hi-C Data" be accepted for publication in PLOS Computational Biology.

All the best,

Eli Zunder

Reviewer's Responses to Questions

**Comments to the Authors:**

Reviewer #1: The authors have addressed all my previous concerns. I suggest to accept this article.

Reviewer #2: The revised manuscript has addressed all my questions.

**Have the authors made all data and (if applicable) computational code underlying the findings in their manuscript fully available?**

Reviewer #1: Yes

Reviewer #2: None

PLOS authors have the option to publish the peer review history of their article (what does this mean?). If published, this will include your full peer review and any attached files.

Reviewer #1: No

Reviewer #2: No

---

## [Editor Report · Acceptance letter]

31 May 2022

PCOMPBIOL-D-21-01981R1 

HiCImpute: A Bayesian Hierarchical Model for Identifying Structural Zeros and Enhancing Single Cell Hi-C Data

Dear Dr Lin,

I am pleased to inform you that your manuscript has been formally accepted for publication in PLOS Computational Biology. Your manuscript is now with our production department and you will be notified of the publication date in due course.

With kind regards,

Andrea Szabo
